# UMoE: Unifying Attention and FFN with Shared Experts

**Yuanhang Yang[1], Chaozheng Wang[2], Jing Li[3]**

[1]Institute of Science Tokyo, Tokyo, Japan
[2]The Chinese University of Hong Kong, Hong Kong, China
[3]Hong Kong Polytechnic University, Hong Kong, China

```
yang.y.ea2c@m.isct.ac.jp
czwang23@cse.cuhk.edu.hk
jing-amelia.li@polyu.edu.hk
```

## Abstract

Sparse Mixture of Experts (MoE) architectures have emerged as a promising approach for scaling Transformer models. While initial works primarily incorporated MoE into feed-forward network (FFN) layers, recent studies have explored extending the MoE paradigm to attention layers to enhance model performance. However, existing attention-based MoE layers require specialized implementations and demonstrate suboptimal performance compared to their FFN-based counterparts. In this paper, we aim to unify MoE designs in attention and FFN layers by introducing a novel reformulation of the attention mechanism, that reveals an underlying FFN-like structure within attention modules. Our proposed architecture, UMoE, achieves superior performance through attention-based MoE layers while enabling efficient parameter sharing between FFN and attention components.

## 1 Introduction

Scaling plays a crucial role in advancing the capabilities of large language models [1, 2, 3]. However, this scaling advantage comes with substantial computational costs, making continued scaling increasingly impractical. Sparse Mixture-of-Experts (MoE) architectures have emerged as a promising solution by selectively activating only a subset of model parameters—termed experts—for each input [4, 5, 6, 7]. This approach effectively decouples model size from computational cost, enabling efficient scaling with minimal overhead.

Recent work has demonstrated the effectiveness of MoE in Transformer architectures [8, 9, 10, 5, 11], particularly when applied to feed-forward neural network (FFN) layers. Building on this success, several studies have explored extending MoE to attention layers [12, 13, 14], indicating potential for performance gains through attention scaling. Despite the potential, we find that existing MoE attention layers demonstrate suboptimal performance compared to FFN-MoE approaches, when provided with similar computational and parametric budgets. This performance gap challenges the practical utility of attention-MoE architectures, as parameters allocated to scaling attention layers might be more effectively utilized for scaling FFNs instead.

We identify two distinctions between attention-MoE and FFN-MoE implementations that likely account for the observed performance differential: (1) the different expert design between attention and FFN layers, and (2) attention-MoE's necessity to compromise the expressiveness of vanilla attention mechanisms to accommodate sparse computation [12]. Motivated by these observations, we investigate a compelling question: can we reformulate attention to reveal an underlying structure compatible with the same expert design as FFN layers, without compromising the expressive power

39th Conference on Neural Information Processing Systems (NeurIPS 2025).

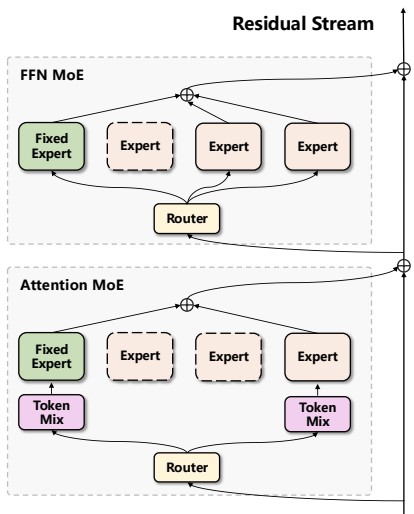

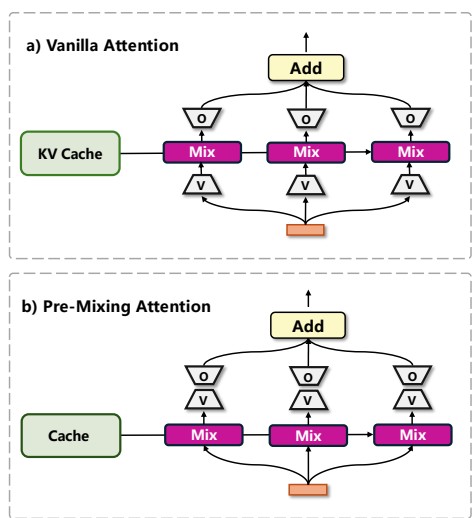

Figure 1: Illustration of a UMoE layer, which incorporates MoE into both FFN and attention modules with shared experts. The primary distinction between attention-MoE and FFN-MoE lies in an additional token mixing operation.

Figure 2: Two formulations of the multi-head attention mechanism. (a) Vanilla attention interleaves mixing operations with value and output projections. (b) Pre-mixing attention performs token mixing prior to projections.

of the attention mechanism? This is a challenging question due to the inherent complexity of attention mechanisms, including multiple projections and softmax calculations, which fundamentally differ from the straightforward two-matrix multiplication pattern of FFNs.

To bridge this structural gap, we reformulate the attention mechanism to reveal its underlying FFN-like structure. Our reformulation decomposes attention into two sequential operations: token mixing and token-wise expert processing. The token-wise expert processing, consisting of two consecutive matrix multiplications, can be implemented as an FFN with a small intermediate size. This implementation naturally aligns with recent advances in fine-grained FFN expert design [6, 15, 3], enabling unified expert architectures and parameter sharing across both attention and FFN layers.

Based on this insight, we introduce UMoE, a unified MoE architecture that abstracts Transformer layers into three fundamental components: *experts*, *token mixing operations*, and *routers*, as shown in Fig. 1. The experts, implemented as standard two-layer FFNs, serve as the primary components for token processing and knowledge storage. The token mixing operations facilitate contextual information exchange through weighted summation of tokens. Routers are employed to dynamically dispatch tokens to the most relevant experts to enable sparse computation. In UMoE, the distinction between the FFN and attention layers lies solely in the expert inputs: FFN layers process tokens independently, while attention layers process tokens simultaneously through weighted summation. This unified design not only simplifies the architecture but also enables parameter-efficient scaling through expert sharing between attention and FFN components.

To evaluate the effectiveness of UMoE, we conduct extensive experiments across various model sizes and tasks, including pre-training and zero-shot evaluations. With the reformulated attention mechanism, the attention-based MoE layers of UMoE match or exceed the performance of previous FFN-based MoE layers. Moreover, by sharing parameters across attention and FFN modules, UMoE achieves superior performance in fully MoE architectures while maintaining the same parameter count. We also present a detailed routing analysis of UMoE, revealing expert specialization patterns across modules, with higher-ranked experts demonstrating interpretable attention patterns. Our code is available at https://github.com/ysngki/UMoE.

## 2 Related Work

**Sparse Mixture-of-Experts (MoE).** Sparse Mixture-of-Experts (MoE) models have gained increasing attention for their ability to scale model capacity while maintaining computational effi-

ciency [4, 9, 10, 5, 11]. The core component of these models is the sparsely activated MoE sub-layer, which selectively activates different parameter subsets for different inputs. In recent Transformer-based implementations, MoE architectures primarily replace feed-forward network (FFN) layers with MoE sub-layers. Each MoE layer consists of a collection of experts, denoted as $\{E_i\}_{i=1}^{N}$, where each expert $E_i$ is implemented as an FFN. Tokens are routed to a subset of experts through a routing mechanism, with the top-k router [4] being the most prevalent approach. Despite advances in routing mechanisms [16, 17, 18, 19, 15], the top-k router remains widely adopted due to its simplicity and robust performance [9]. For a given token $x \in \mathbb{R}^d$, where $d$ is the hidden dimension, and a trainable weight matrix $\mathbf{W}_r \in \mathbb{R}^{N \times d}$, the top-k router computes the probability distribution over experts as:

$$p = \texttt{softmax}(\mathbf{W}_r x). \tag{1}$$

The set of top-$k$ experts $\mathcal{T}$ is then selected based on $p$, where $|\mathcal{T}| = k$. Each expert processes the token independently and the final output of the MoE layer is computed as the weighted combination of these $k$ experts' outputs:

$$y = \sum_{i \in \mathcal{T}} p_i E_i(x), \tag{2}$$

where each expert is implemented as an FFN with two matrices and a non-linear activation function.

**MoE for Attention.** Several recent approaches have explored extending the MoE paradigm to attention layers in Transformers [12, 13], with a primary focus on expert design. Because attention layers lack the consecutive matrix multiplication pattern found in FFNs, these approaches necessitate expert designs that differ from FFN-MoE models. The Mixture-of-Attention (MoA) [12] propose to conceptualize individual attention heads as experts, scaling attention layers by increasing the number of attention heads. However, introducing sparsity into attention layers presents a significant challenge: query vectors computed by a specific expert (or head) require corresponding key and value vectors from the same expert, necessitating identical expert activation across all tokens. To address this constraint, MoA implements distinct query and output projections per head while maintaining shared key and value projections across attention heads.

SwitchHead [13] presents an alternative approach to implementing the MoE paradigm in attention layers. Rather than treating entire attention heads as experts, SwitchHead designates individual projection matrices within heads as experts. A straightforward implementation maintains four separate MoE sub-layers per head for query, key, value, and output projections. While scaling all projections yields performance improvements, empirical results show that value and output projections benefit most significantly from scaling.

In contrast to these approaches, UMoE unifies attention-MoE and FFN-MoE through a novel reformulation of the multi-head attention mechanism, enabling the shared expert design and parameters across both attention and FFN layers.

**Other Related Work.** Several studies have explored connections between MoE and attention from different perspectives. MoH [20] proposes using MoE for pruning attention heads in LLMs by continuing pre-training with a routing function. During inference, certain output projections ($W_o$), viewed as experts, are selectively skipped based on routing decisions. Taking a different approach, MH-MoE [21] incorporates concepts from multi-head attention to enhance FFN-based MoE models. Instead of routing original input tokens to experts, MH-MoE decomposes each token into multiple low-dimensional sub-tokens, which are then processed in parallel by diverse sets of experts.

## 3 Method

The attention mechanism is the core of Transformers [22], processing token hidden states to capture contextual relationships. However, its structure differs from FFN layers, which complicates the unification of MoE designs across both modules. In this section, we present two alternative formulations of attention, pre-mixing and post-mixing, that reveal an inherent FFN-like structure within attention layers. Based on these formulations, we introduce a novel MoE architecture, UMoE.

```
def UMoELayer(x, X):
    # x: [1, d], X: [n, d]

    ### Attention MoE
    indices, probs = TopKRouter(x) # Assign token x to Experts

    residual_x = x.copy()
    K = X @ W_k
    q_shared = x @ W_q
    for i, p in zip(indices, probs):
        q = q_shared + x @ W_a[i] @ W_b[i]
        # K and V (the hidden states X) are shared across experts.
        y = Attention(Q=q, K=K, V=X)
        residual_x += p * Experts[i](y)
    x = residual_x

    ### FFN MoE
    indices, probs = TopKRouter(x) # Assign token x to Experts
    residual_x = x.copy()
    for i, p in zip(indices, probs):
        residual_x += p * Experts[i](x)
    return residual_x
```

Figure 3: Implementation details of a UMoE layer. The input consists of a sequence X containing n token hidden states and x representing the final hidden state. For simplicity, this implementation focuses on computing the output for the last token.

### 3.1 Formulations of Attention

**Preliminaries.** Consider a sequence of token hidden states $\mathbf{X} \in \mathbb{R}^{n \times d}$, where $n$ is the sequence length and $d$ is the hidden dimension. In multi-head attention, each token attends to all other tokens in the sequence through query, key, and value projections. For a single token $\boldsymbol{x}$ (e.g., the last token in the sequence for simplicity), its attention output is computed as:

$$\boldsymbol{q} = \boldsymbol{x}\mathbf{W}_q, \ \mathbf{K} = \mathbf{X}\mathbf{W}_k, \ \mathbf{V} = \mathbf{X}\mathbf{W}_v, \tag{3}$$

$$\boldsymbol{a} = \texttt{softmax}\left(\frac{\boldsymbol{q}\mathbf{K}^\top}{\sqrt{d_k}}\right), \quad \boldsymbol{o} = \boldsymbol{a}\mathbf{V}, \tag{4}$$

where $\mathbf{W}_q, \mathbf{W}_k \in \mathbb{R}^{d \times d_k}$ and $\mathbf{W}_v \in \mathbb{R}^{d \times d_v}$ are learnable matrices, respectively, and $\boldsymbol{a} \in \mathbb{R}^n$ is the attention weight. To enhance representation capacity, this process is repeated $h$ times in parallel, and the outputs are combined:

$$\boldsymbol{y} = [\boldsymbol{o}_1; \boldsymbol{o}_2; \cdots ; \boldsymbol{o}_h]\mathbf{W}_o, \tag{5}$$

where $\mathbf{W}_o \in \mathbb{R}^{hd_v \times d}$ projects the concatenated outputs back to the original dimension $d$.

**Pre-Mixing Formulation.** While multi-head attention is typically expressed using concatenation, it can be equivalently expressed as a sum of per-head outputs, which helps reveal its connection to FFN layers. By decomposing $\mathbf{W}_o$ into small matrices $\mathbf{W}_o^i \in \mathbb{R}^{d_v \times d}$ along the feature dimension, we can express the output as:

$$\boldsymbol{y} = \sum_{i=1}^{h} \boldsymbol{o}_i \mathbf{W}_o^i \quad = \sum_{i=1}^{h} (\boldsymbol{a}_i \mathbf{X} \mathbf{W}_v^i) \mathbf{W}_o^i \tag{6}$$

$$= \sum_{i=1}^{h} (\boldsymbol{a}_i \mathbf{X})(\mathbf{W}_v^i \mathbf{W}_o^i). \tag{7}$$

This reformulation provides two distinct interpretations, as shown in Fig. 2:

- Eq. 6: The conventional view where value vectors are first aggregated then projected back into the hidden space with an output projection.

- Eq. 7: A new interpretation where token hidden states are first aggregated into contextualized representations, i.e., weighted averages of all tokens, before being processed by the value ($\mathbf{W}_v^i$) and output ($\mathbf{W}_o^i$) projections. We term this formulation as *pre-mixing* attention.

While both interpretations yield same outputs, the pre-mixing formulation enables the grouping of $W_o$ and $W_v$. This grouping reveals that pre-mixing attention exhibits a two-layer structure analogous to FFN modules, which can be implemented as a linear FFN with no activation function.

**Post-Mixing Formulation.** Alternatively, we can rearrange the computation as:

$$\boldsymbol{y} = \sum_{i=1}^{h} \boldsymbol{a}_i(\mathbf{X}\mathbf{W}_v^i\mathbf{W}_o^i). \tag{8}$$

In this formulation, token hidden states are transformed by two successive projections independently for each token, before being aggregated using the attention weights.

## 3.2 UMoE

By grouping $\mathbf{W}_v$ and $\mathbf{W}_o$, both pre-mixing and post-mixing attention can be naturally interpreted as a MoE architecture, aligning with established FFN-MoE practices. Using pre-mixing attention as an example, let the expert $\mathrm{E}(\boldsymbol{x}) := \boldsymbol{x}\mathbf{W}_v\mathbf{W}_o$. The multi-head attention can then be reformulated as:

$$\boldsymbol{y} = \sum_{i=1}^{h} \mathrm{E}_i(\boldsymbol{a}_i\mathbf{X}). \tag{9}$$

By increasing the number of experts and introducing a routing mechanism, such as a `top-k` router, we derive a MoE architecture, denoted as UMoE-Att. The output of a UMoE-Att layer is:

$$\boldsymbol{y} = \sum_{i\in\mathcal{T}} \boldsymbol{p}_i\mathrm{E}_i(\boldsymbol{a}_i\mathbf{X}), \quad \text{where } \mathcal{T} \text{ is the set of activated experts.} \tag{10}$$

Referring to Eq. 2, we observe that the primary distinction between FFN-MoE layers and UMoE-Att layers lies in their expert inputs: FFN experts operate on individual token hidden states $\boldsymbol{x}$, while attention experts process weighted combinations of all token hidden states. This reveals a relationship: FFN-MoE layers can be interpreted as a specialized case of pre-mixing attention layers where the attention matrix is constrained to an identity matrix, limiting each token to self-attention only.

**Fully MoE Architecture.** Both the experts in UMoE-Att and the FFN layers of Transformer consist of two consecutive matrices. While attention layer experts utilize a relatively small intermediate size ($d_v$), FFN layers typically employ larger dimensions. Recent advances in FFN-MoE models suggest the efficacy of using FFN layers with reduced intermediate sizes as experts [6, 15, 3]. This insight enables the direct adoption of experts in attention layers for FFN layers, resulting in a fully MoE architecture, denoted as UMoE. Fig. 1 illustrates the architecture of a UMoE layer., where the MoE paradigm is applied to both FFN and attention layers using a shared expert set. Notably, to facilitate parameter sharing, experts are implemented as two-layer FFNs with an intermediate size of $d_v$ and incorporate a non-linear activation function between matrix multiplications.

**Pre-mixing Implementation.** The token mixing operation in pre-mixing attention is a weighted summation over token hidden states, which can be implemented as vanilla attention, accepting $Q$, $K$, $V$ matrices as input and producing an output matrix. Each token generates distinct query vectors for different experts, while values (hidden states) and their associated keys are shared across experts. To generate expert-dependent queries for input tokens, each expert requires an additional query projection matrix, leading to a notable increase in parameters. To mitigate the parameter count disparity with existing MoE models, where experts typically comprise two matrices, we employ low-rank matrices [23] for query projection within UMoE experts. For a given token $\boldsymbol{x}$, the query for expert $i$ is computed as:

$$\boldsymbol{q}_i = \boldsymbol{x}\mathbf{W}_q + \boldsymbol{x}\mathbf{W}_a^i\mathbf{W}_b^i, \tag{11}$$

where the first term is shared across all experts, while the second term is expert-specific with unique parameters, $\mathbf{W}_a^i \in \mathbb{R}^{d\times r}$ and $\mathbf{W}_b^i \in \mathbb{R}^{r\times d_k}$, for each expert. Fig. 3 presents the pseudo-code of a UMoE layer.

**Putting It All Together.** As illustrated in Fig. 1, UMoE integrates three key components: (1) experts implemented as fine-grained FFNs with dual low-rank query projection matrices, (2) pre-mixing attention mechanism utilizing shared keys and values across experts, and (3) the `top-k` router for expert selection. It is noteworthy that while MoA [12] also shares keys and values across experts, the values of MoA are the results after applying a value linear transformation to the input token hidden states. In contrast, the values of UMoE directly refer to the input token hidden states.

## 3.3 Discussion

**Vanilla Attention vs Pre-mixing Attention.** Vanilla attention and pre-mixing attention differ in terms of KV cache requirements and computational complexity. During inference, vanilla attention requires caching multiple keys and values per token, whereas pre-mixing attention requires only one key and hidden state per token. While grouped-query attention (GQA) is commonly adopted to reduce the KV cache size in attention layers, it cannot be combined with pre-mixing attention, since the latter already maintains only a single key–value pair per token. Instead, multi-head latent attention (MLA) [8], a promising alternative to GQA, can be applied by introducing a down-projection to the hidden states before the token-mixing operation. Regarding the computation, while vanilla attention performs weighted summation over low-dimensional value vectors, pre-mixing attention operates on input token hidden states, introducing a modest increase in computational complexity. This modest increase, however, becomes increasingly negligible as models scale to larger dimensions, effectively amortizing the additional computational overhead. A detailed comparative analysis is presented in Table 8 (A.1). Additionally, the abstract formulation of UMoE opens avenues for future research to explore more computationally efficient token mixing alternatives, such as linear attention mechanisms [24, 25].

**Pre-mixing Attention vs Post-mixing Attention.** As illustrated in Fig. 4, post-mixing attention processes individual tokens through experts prior to mixing. The architectural distinction between pre-mixing and post-mixing variants represents different perspectives on token-parameter interactions in attention layers. Recent interpretability studies have drawn parallels between the two-matrix multiplication pattern of FFNs and associative memory modules, where value neurons, i.e. columns of the second matrix in FFNs, are retrieved by inputs [26, 27, 28]. Within this framework, pre-mixing attention leverages token mix-

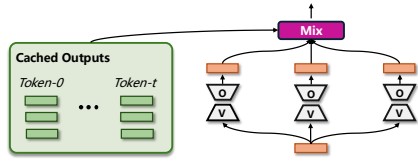

Figure 4: Post-Mixing Attention.

ing to generate contextualized inputs for precise retrieval. In contrast, post-mixing attention can be conceptualized as an ensemble of independent retrievals executed by preceding tokens. Our preliminary experiments (A.2) demonstrates a significant performance advantage of pre-mixing attention over its post-mixing counterpart. This observation suggests that generating contextualized inputs for token-parameter interactions more effectively aligns with the principles of attention mechanisms.

# 4 Experiments

## 4.1 Setup

**Datasets.** We conduct language modeling pretraining on two datasets: FineWeb-Edu 100B [29] and Wikitext-103 [30]. FineWeb-Edu has shown superior data efficiency when evaluated on knowledge-intensive benchmarks. Wikitext-103, consisting of approximately 100M tokens, is a smaller corpus that has been widely adopted in previous studies [16, 13]. We apply the LLaMA tokenizer [31] with a 32K vocabulary size to both datasets. The zero-shot performance of models trained on FineWeb-Edu is evaluated using the `lm-evaluation-harness` framework [32].

**Baselines.** We compare UMoE against three categories of baselines: dense models, FFN-MoE models with fine-grained experts [8, 3], and attention-MoE models (specifically MoA [12] and SwitchHead [13]). Attention-MoE models are configured with identical expert parameters[1]. We

---

[1]SwitchHead represents an exception, as it requires the number of experts to be divisible by the number of attention heads.

Table 1: Comparison of Dense and Sparse Mixture-of-Experts (MoE) Models for Language Modeling. In MoE models, $A \times B$ denotes $B$ experts per layer with size $A$ in '#Total' columns, while $B$ in '#Active' columns indicates the number of experts activated per token. Gray entries in '#Total' columns indicate shared parameters between attention and FFN modules in UMoE models. UMoE-Att refers to UMoE variants with MoE only applied to attention modules.

| Model | Params | Attention | | FFN | | PPL (↓) | | MACs |
|---|---|---|---|---|---|---|---|---|
| | | #Total | #Active | #Total | #Active | Fineweb | Wikitext | |
| *Base Models* | | | | | | | | |
| Dense | 134 M | 768 | 768 | 3072 | 3072 | 25.79 | 30.41 | 525 G |
| Fine-grained FFN-MoE | 535 M | 768 | 768 | $192{\times}128$ | $192{\times}16$ | 21.19 | 27.94 | 530 G |
| MoA | 525 M | $192{\times}116$ | $192{\times}4$ | 3072 | 3072 | 22.28 | 27.57 | 486 G |
| SwitchHead | 533 M | $192{\times}116$ | $192{\times}4$ | 3072 | 3072 | 22.91 | 29.47 | 542 G |
| UMoE-Att | 547 M | $192{\times}116$ | $192{\times}4$ | 3072 | 3072 | 20.81 | 27.45 | 611 G |
| UMoE | 540 M | $192{\times}128$ | $192{\times}4$ | $192{\times}128$ | $192{\times}16$ | **20.44** | **26.67** | 616 G |
| *Large Models* | | | | | | | | |
| Dense | 1.1 B | 2048 | 2048 | 5632 | 5632 | 17.53 | 25.46 | 4.59 T |
| Fine-grained FFN-MoE | 3.8 B | 2048 | 2048 | $512{\times}64$ | $512{\times}11$ | 16.09 | 25.47 | 4.61 T |
| MoA | 3.6 B | $512{\times}57$ | $512{\times}4$ | 5632 | 5632 | 16.72 | **25.14** | 3.99 T |
| SwitchHead | 3.7 B | $512{\times}60$ | $512{\times}4$ | 5632 | 5632 | 16.48 | 27.24 | 4.62 T |
| UMoE-Att | 3.8 B | $512{\times}57$ | $512{\times}4$ | 5632 | 5632 | 16.03 | 25.53 | 4.73 T |
| UMoE | 3.6 B | $512{\times}64$ | $512{\times}4$ | $512{\times}64$ | $512{\times}11$ | **15.95** | 25.44 | 4.75 T |

implement all MoE models with a fixed expert per layer, following recommendations for optimal model performance [8, 33]. UMoE adopts the pre-mixing attention mechanism, which consistently outperforms post-mixing variants.

**Experimental Setup.** UMoE is implemented as a decoder-only Transformer with rotary position embedding [34], following deepseek-MoE [6]. The load balancing loss proposed by Switch Transformer [5] is adopted to encourage a balanced load across experts. The experts in our experiments are implemented as two-layer MLPs to ensure fair comparison with the baselines.[2]

We evaluate two model configurations: The base models comprise 12 layers with a hidden size of 768, while the large models consist of 24 layers with a hidden size of 2048. These configurations yield dense models with 134M and 1.1B parameters, respectively. MoE variants replace all attention or FFN layers with MoE layers. Due to computational constraints, unless otherwise specified, models are pretrained on 50B tokens from FineWeb-Edu with batch size of 1024. For Wikitext-103, following Csordás et al. [13], models are trained for 100k steps, though models typically overfit within 20k steps. Detailed hyperparameters are provided in A.4.

## 4.2 Comparison with Baselines

**Results.** From Table 1, we observe that UMoE shows consistent superiority across different model sizes and datasets. In the base model regime, UMoE achieves the best performance. Notably, the attention-only variant of UMoE exhibits substantial improvements over previous attention-based approaches. Even without parameter sharing, UMoE-Att establishes itself as a compelling alternative to traditional FFN-based MoE models. The parameter sharing mechanism between attention and FFN modules further enhances the effectiveness without increasing the total parameter count. Despite equalizing the number of activated experts across all baselines, we observe subtle computational discrepancies due to different attention layer implementations, as measured by MACs (multiplication accumulation operation). We additionally perform a MAC-matched comparison by increasing the number of activated experts of baseline models. Table 2 shows that even under comparable computational constraints, UMoE method achieves the lowest perplexity.

---

[2]UMoE is designed as a flexible framework agnostic to the expert implementation. Using more advanced gated MLP variants, such as SwiGLU, may lead to improved performance, as the gating mechanism can enhance the expressiveness of both attention and feed-forward layers.

Table 2: MAC-matched comparison for base models by increasing the number of activated experts of baseline models.

| Model | MACs | Active Params | PPL (↓) | |
|---|---|---|---|---|
| | | | Fineweb | Wikitext |
| FFN-MoE | 617 G | $768 + 192 \times 22$ | 20.80 | 27.39 |
| MoA | 621 G | $192 \times 8 + 3072$ | 22.00 | 27.63 |
| SwitchHead | 649 G | $192 \times 12 + 3072$ | 21.57 | 28.13 |
| UMoE-Att | 611 G | $192 \times 4 + 3072$ | 20.81 | 27.45 |
| UMoE | 616 G | $192 \times 4 + 192 \times 16$ | __20.44__ | __26.67__ |

Table 3: Parameter sharing strategies. ✓ indicates shared components between modules while ✗ indicates separate components.

| Component | UMoE | - | - | - |
|---|---|---|---|---|
| Fixed Experts | ✗ | ✓ | ✗ | ✓ |
| Router | ✗ | ✓ | ✓ | ✗ |
| # Params | 540 M | 536 M | 540 M | 537 M |
| PPL | 22.82 | 23.11 | 23.05 | 23.02 |

Table 4: Zero-shot accuracy on downstream tasks. The best score is marked in **bold**.

| Model | Params | HellaSwag | PIQA | ARC-E | ARC-C | RACE | Lambada | MMLU | Wino | Avg. |
|---|---|---|---|---|---|---|---|---|---|---|
| *Base Models* | | | | | | | | | | |
| Dense | 134 M | 33.58 | 62.35 | 46.09 | 24.74 | 27.75 | 19.97 | 24.8 | 49.8 | 36.14 |
| MoA | 525 M | 37.82 | 65.58 | 51.34 | 26.19 | 28.83 | 22.33 | 25.1 | 50.7 | 38.49 |
| SwitchHead | 533 M | 37.19 | 66.12 | 50.55 | 26.59 | 28.14 | 21.73 | 25.2 | 50.9 | 38.30 |
| FFN-MoE | 535 M | 39.69 | 66.43 | **52.95** | 26.71 | **29.76** | 23.46 | 25.3 | 52.1 | 39.55 |
| UMoE (Att) | 547 M | 40.72 | **67.36** | 51.77 | 27.82 | **29.76** | 23.66 | 25.9 | 52.5 | 39.94 |
| UMoE | 540 M | **41.28** | 66.65 | 51.86 | **29.01** | 28.71 | **23.77** | 26.6 | 52.6 | __40.06__ |
| *Large Models* | | | | | | | | | | |
| Dense | 1.1 B | 48.45 | 69.26 | 58.85 | 32.17 | 33.11 | 31.75 | 27.4 | 53.8 | 44.35 |
| MoA | 3.6 B | 50.61 | 70.28 | 61.47 | 33.22 | 32.38 | 33.15 | 28.6 | 54.7 | 45.55 |
| SwitchHead | 3.7 B | 51.90 | 70.83 | 62.34 | 33.69 | 33.27 | 33.66 | 28.8 | 55.7 | 46.27 |
| FFN-MoE | 3.8 B | 52.74 | 71.52 | **64.23** | 35.67 | **33.30** | 34.00 | 29.2 | 56.3 | 47.12 |
| UMoE (Att) | 3.8 B | **53.20** | 71.44 | 63.30 | 34.39 | 32.82 | 34.78 | 29.3 | **57.4** | 47.08 |
| UMoE | 3.6 B | 53.17 | **72.47** | **64.23** | 35.75 | 32.44 | **35.32** | 30.4 | 56.9 | __47.58__ |

In larger-scale models, UMoE maintains its competitive advantage. While MoA shows marginally better performance on Wikitext-103, this result may not fully reflect model capabilities given the relatively small size of Wikitext-103 (100M tokens) compared to the model scale. Following established practice [16], we further report validation perplexity. Fig. 5 shows that UMoE demonstrates faster convergence and lower validation perplexity compared to baselines, indicating enhanced modeling capabilities. This superior performance translates to downstream tasks, with Table 4 showing UMoE consistently achieving the highest average zero-shot accuracy across diverse tasks.

**Efficiency.** Following Zhang et al. [12], Jin et al. [20], we employ MACs[3] as an efficiency metric, as it remains independent of hardware implementations. As shown in Table 1, the pre-mixing attention introduces a modest computational overhead, resulting in approximately 1.17× slowdown for base models. However, this slowdown becomes increasingly negligible as models scale up; in large models, UMoE introduces only 1.03× slowdown compared to the dense baseline. This favorable scaling behavior arises from the different growth rates in computational complexity: expert processing scales quadratically with hidden dimension, while token aggregation in attention layers scales linearly.

### 4.3 Ablations

We conducted ablation experiments using base models trained on FineWeb-Edu with 20B tokens.

**Parameter Sharing Analysis.** We investigated various sharing strategies across FFN and attention layers for fixed experts [6, 33] and routers. As shown in Table 3, all configurations achieved comparable perplexity. Our default configuration, which employs separate fixed experts and routers across FFN and attention layers, yielded the optimal perplexity.

**Expert Allocation.** As suggested in Section 3.2, FFN-MoE layers can be interpreted as a specialized case of pre-mixing attention layers with an identity matrix as attention matrix. This interpretation raises a question: *does UMoE perform better when allocating more experts to attention layers rather than FFN layers?* According to Table 5, we observe an trend when gradually shifting expert

---

[3]MACs is measured using the DeepSpeed Flops Profiler.

Table 5: Impact of expert allocation between Attention and FFN layers (total experts = 20).

| Model | # Expert | | PPL |
|---|---|---|---|
| | Attention | FFN | |
| UMoE | 4 | 16 | 22.82 |
| | 8 | 12 | 22.63 |
| | 12 | 8 | 22.44 |
| | 16 | 4 | 22.50 |
| | 20 | 0 | 21.75 |

Table 6: Effect of Activation Functions in Expert Modules. ✓indicates experts with activation functions while ✗ indicates experts without activation functions.

| Model | Act. Function | PPL |
|---|---|---|
| UMoE | ✓ | 22.82 |
| | ✗ | 24.43 |
| UMoE (Att) | ✓ | 23.37 |
| | ✗ | 23.99 |

Table 7: Top tokens for selected experts in the last attention and FFN layer of UMoE.

| Expert ID | Top Tokens in Attention Layer | Top Tokens in FFN Layer |
|---|---|---|
| 3 | _Each , This , Every , Each , _This | This , _This , Every , _Each , _Another |
| 10 | _Film , _video , _lab , _film , _Video | Tag , _Font , _ISBN , twitter , _DNS |
| 46 | The , _The , _the , the , _Our | _a , _his , _my , Your , _Your |
| 64 | ” , %. , ." , :) , ." | _relatively , _extremely , _a , _very , _Very |

allocation from FFN to attention modules while maintaining a total activated expert size of 20. The model achieves its best perplexity when all experts are allocated to attention layers. This finding provides empirical evidence supporting our theoretical interpretation that FFN layers function as a specialized form of attention, with the attention mechanism exhibiting greater expressiveness. However, increasing the number of attention experts introduces substantial computational overhead due to token mixing operations. Future research could explore efficient attention alternatives within the attention MoE framework.

**Activation Function.** Table 6 presents our investigation into the impact of activation functions in UMoE. The results demonstrate that incorporating activation functions between matrix multiplications within experts consistently improves model performance, reinforcing the crucial role of non-linearity in deep learning architectures. Notably, while the removal of activation functions reduces the experts to pure linear transformations in both FFN and attention modules, UMoE remains trainable. We attribute this robustness to the preserved non-linearity from token mixing operations and layer normalization. Nevertheless, the consistent performance degradation underscores the importance of activation functions in model expressiveness, particularly in the context of shared expert architectures.

### 4.4 Expert Specialization

Table 7 presents the routing patterns in the final layer of UMoE, where experts are shared between attention and FFN modules while maintaining distinct routers. Notably, certain token categories consistently route to the same experts across both modules, as evidenced by experts 3 and 46. Expert 3 consistently processes determiners, while expert 46 specializes in demonstrative pronouns.

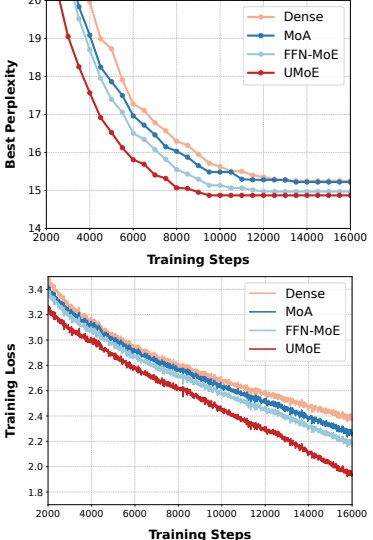

Figure 5: Best valid PPL (top) and training loss (bottom) on Wikitext.

The analysis also reveals divergent specialization patterns that highlight the complexity of shared expert architectures. A notable example is expert 64, which exhibits distinct specializations: processing consecutive punctuation marks in the attention layer while handling degree adverbs in the FFN layer. This phenomenon suggests that shared experts can develop multiple specializations, potentially leading to more efficient parameter utilization. However,

it also raises important questions about potential knowledge conflicts within individual experts, indicating promising directions for future research in routing mechanism design for shared expert architectures.

We also provide an analysis on the attention maps of UMoE in A.5, which confirms that higher-ranked experts show more focused attention distributions on relevant tokens compared to lower-ranked ones.

## 5 Conclusion

The paper proposes UMoE, a novel architecture that unifies MoE designs for attention and FFN layers. The key insight is a reformulation of the attention mechanism that allows the value and output projections to be grouped into FFN-like experts. This unification enables parameter sharing across attention and FFN layers, resulting in a fully MoE architecture that improves performance without introducing additional parameters. The paper presents extensive experiments demonstrating UMoE 's superiority over existing MoE architectures in terms of perplexity on language modeling datasets and accuracy on zero-shot tasks.

As for future work, we are looking at replacing the token mixing mechanism with more efficient alternatives to enable scaling up the number of activated experts in attention layers. In addition, we are also interested in investigating architectures that unify attention and FFN into a single layer, given our finding that FFN layers function as a specialized case of attention with reduced expressiveness. Exploring parameter sharing across different Transformer layers [35, 36] is also a promising direction.

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

Table 8: Computational complexity analysis of vanilla attention and pre-mixing attention mechanisms. Entries in gray denote identical complexity terms between the two mechanisms. Here, $N$ denotes sequence length, $d$ represents hidden dimension, $h$ indicates the number of attention heads (activated experts), $d_k$ and $d_v$ are key and value dimensions respectively, and $r$ is the rank of query projection matrices in the pre-mixing attention.

| Operation | Vanilla | Pre-mixing |
|---|---|---|
| Output Projection | $\mathcal{O}(N \cdot d_v \cdot d \cdot h)$ | $\mathcal{O}(N \cdot d_v \cdot d \cdot h)$ |
| Value Projection | $\mathcal{O}(N \cdot d_v \cdot d \cdot h)$ | $\mathcal{O}(N \cdot d_v \cdot d \cdot h)$ |
| Key Projection | $\mathcal{O}(N \cdot d_k \cdot d \cdot h)$ | $\mathcal{O}(N \cdot d_k \cdot d)$ |
| Query Projection | $\mathcal{O}(N \cdot d_k \cdot d \cdot h)$ | $\mathcal{O}(N \cdot d_k \cdot d + N \cdot (d_k + d) \cdot r \cdot h)$ |
| QK Multiplication | $\mathcal{O}(N^2 \cdot d_k \cdot h)$ | $\mathcal{O}(N^2 \cdot d_k \cdot h)$ |
| Weighted Sum | $\mathcal{O}(N^2 \cdot d_v \cdot h)$ | $\mathcal{O}(N^2 \cdot d \cdot h)$ |

# A  Technical Appendices and Supplementary Material

## A.1  Complexity

Table 8 presents a detailed computational complexity comparison between vanilla attention and pre-mixing attention mechanisms. The key distinctions lies in two operations: key projection and weighted sum computation. In key projection, pre-mixing attention achieves lower complexity. However, this efficiency is partially offset in the weighted sum operation, where the complexity increases due to the full dimensional mixing. It is worth mentioning that the computational complexity of weighted sum grows linearly with the hidden dimension, while the FFN computation grows quadratically. The computational overhead of weighted sum becomes less significant as model size increases.

```
def PostMixingMoE(X):
    # X: [n, d]

    ### Attention MoE
    # Independent Expert Processing
    all_token_outputs = []
    all_token_keys = []

    for this_token in X:
        indices, probs = TopKRouter(this_token)

        for i, p in zip(indices, probs):
            y = p * Experts[i](this_token)
            all_token_outputs.append(y)

            k = W_k[i](this_token)
            all_token_keys.append(k)

    # Mixing
    Q = (X @ W_q).unsuqeeze(1) # [n, 1, d_q]
    K = all_token_keys.reshape(n, k, d_q)
    V = all_token_outputs.reshape(n, k, d_v)

    attn_output = Attention(Q, K, V) # [n, 1, d_v]

    ### FFN MoE
    ...
```

Figure 6: Implementation of a UMoE layer based on post-mixing attention. $X$ is a sequence of $n$ token hidden states. In the attention layer, tokens are processed by their top-$k$ experts independently. The output embeddings of all tokens are aggregated according to the attention weights.

## A.2 Post-mixing vs Pre-mixing

Fig. 6 presents the pseudo-code of a UMoE layer based on post-mixing attention. We conducted preliminary experiments on Wikitext-103 and FineWeb-Edu datasets. As illustrated in Fig. 7, MoE models incorporating pre-mixing attention demonstrate substantially superior performance compared to their post-mixing counterparts.

As discussed in Section 3.1, these two reformulations of the attention mechanism are mathematically equivalent, owing to the absence of non-linear transformations within the matrix chain multiplication in attention heads. However, when grouping the value and output projections and implementing them as an FFN with a non-linear activation function, these formulations yield distinct outputs in the MoE layers. Fig. 8 illustrates MoE models implemented based on these two reformulations. Both approaches can be interpreted as extensions of conventional FFN-based MoE models. Specifically, the pre-mixing approach enhances FFN-based MoE models by contextualizing the inputs, while post-mixing attention enables MoE layers to incorporate other tokens' outputs in generating the final output. Future research could explore the synergistic combination of post- and pre-mixing approaches to fully leverage contextual information.

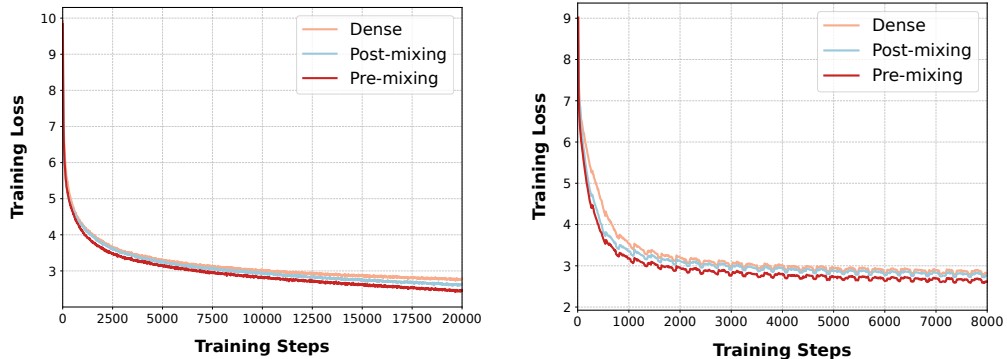

Figure 7: Loss curves of UMoE with attention MoE layers implemented on post-mixing and pre-mixing attention, respectively. Models are trained on Wikitext-103 (left) and FineWeb-Edu (right).

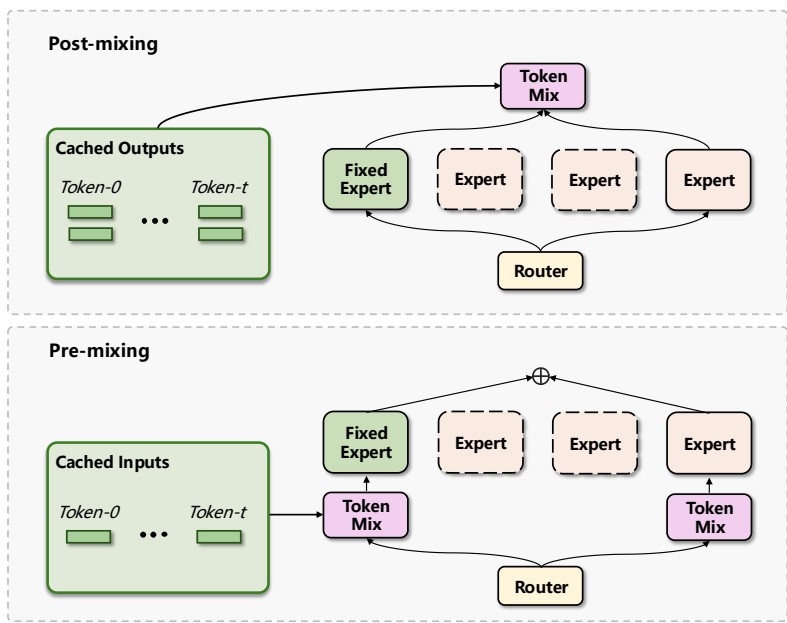

Figure 8: Two implementations of UMoE based on pre-mixing and post-mixing attention, respectively.

Table 9: Comparison of training throughput and inference latency across models. Inference latency denotes the pre-filling time (1024 tokens) measured on a single H100 GPU. All results are averaged over multiple runs.

| Model Type | Model | Model Size | Pre-filling Latency (s) | Training Throughput (tokens/s) |
|---|---|---|---|---|
| **Base Model** | GPT | 134M | **0.1376** | **126,508** |
| | FFN-MoE | 535M | 0.1974 | 74,415 |
| | MoA | 525M | 0.1843 | 90,394 |
| | SwitchHead | 533M | 0.2253 | 69,374 |
| | UMoE-Att | 547M | 0.2006 | 76,538 |
| | UMoE | 540M | 0.2197 | 71,461 |
| **Large Model** | GPT | 1.1B | **0.2950** | 11,634 |
| | FFN-MoE | 3.8B | 0.4080 | 9,799 |
| | MoA | 3.6B | 0.3624 | **12,321** |
| | SwitchHead | 3.7B | 0.4328 | 8,882 |
| | UMoE-Att | 3.8B | 0.4120 | 11,377 |
| | UMoE | 3.6B | 0.4392 | 9,194 |

## A.3 Wall-clock Efficiency Analysis

Table 9 presents a quantitative comparison of end-to-end training throughput and inference latency across various model architectures. All experiments were conducted on a single NVIDIA H100 GPU. Inference latency corresponds to the pre-filling time for sequences of 1024 tokens, while training throughput is measured in tokens per second.

Across both scales, all Mixture-of-Experts (MoE) variants exhibit comparable inference latency but remain slower than the dense baseline, despite their similar theoretical MAC counts. This discrepancy primarily arises from additional computational overheads introduced by expert routing and sparse expert execution. Among MoE models, SwitchHead and UMoE show slightly higher latency due to their architectural designs—SwitchHead applies two MoE layers per attention head, whereas UMoE employs two MoE layers within each Transformer block.

These results underscore a persistent challenge in MoE-based architectures: while their theoretical efficiency is well established, current hardware and software infrastructures (e.g., routing kernels, expert parallelization) are not yet fully optimized to realize these potential speedups. Consequently, our main paper reports MAC-based efficiency as the primary metric. We anticipate that advances in GPU kernel design and distributed parallelization will further mitigate the observed wall-clock inefficiencies in future implementations.

## A.4 Hyperparameters

Table 10 and 11 give the parameters used for based and large models, respectively. It takes roughly a week for pretraining base models on FineWeb-Edu datasets. MoA and SwitchHead utilize identical parameters as UMoE-Att, excluding the low-rank query projections. Table!12 details the hyperparameters used during training. For the Wikitext-103 dataset, we adopt the same hyperparameter configuration as Csordás et al. [13].

## A.5 Attention Analysis

We analyze the attention patterns in UMoE by visualizing expert-specific attention maps. While UMoE (Large) contains 64 experts per layer across 24 layers, we focus on the top 8 experts (ranked by router scores) to maintain tractability. Each expert utilizes its own query projection matrix, allowing us to compute attention maps regardless of activation status.

To investigate attention behavior, we examine two inputs:

- "*Context: William Shakespeare wrote the famous play Romeo and Juliet in the late 16th century. Question: Who wrote Romeo and Juliet? The Answer is*"

Table 10: Hyperparameters of Base Models. MoA and SwitchHead use the same hyperparameters as UMoE-Att.

| Hyperparameter | Dense | FFN-MoE | UMoE-Att | UMoE |
|---|---|---|---|---|
| Context Length | 1024 | 1024 | 1024 | 768 |
| Number of Layers | 12 | 12 | 12 | 12 |
| Hidden Size | 768 | 768 | 768 | 768 |
| Attention Heads | 4 | 4 | 4 | 4 |
| FFN Size | 3072 | $192 \times 16$ | 3072 | $192 \times 16$ |
| Query (Key) Dimension | 512 | 512 | 512 | 512 |
| Value Dimension | 192 | 192 | – | – |
| Query Lora Rank | – | – | 16 | 16 |
| Number of MoE layers | – | 12 | 12 | 12 |
| Expert Size | - | 192 | 192 | 192 |
| Experts per MoE Layer | – | 128 | 116 | 128 |
| FFN Experts per Token | – | 16 | – | 16 |
| Attention Experts per Token | – | – | 4 | 4 |

Table 11: Hyperparameters of Large Models. MoA and SwitchHead use the same hyperparameters as UMoE-Att.

| Hyperparameter | Dense | FFN-MoE | UMoE-Att | UMoE |
|---|---|---|---|---|
| Context Length | 1024 | 1024 | 1024 | 1024 |
| Number of Layers | 24 | 24 | 24 | 24 |
| Hidden Size | 2048 | 2048 | 2048 | 2048 |
| Attention Heads | 4 | 4 | 4 | 4 |
| FFN Size | 5632 | $512 \times 11$ | 5632 | $512 \times 11$ |
| Query (Key) Dimension | 512 | 512 | 512 | 512 |
| Value Dimension | 512 | 512 | – | – |
| Query Lora Rank | – | – | 36 | 36 |
| Number of MoE layers | – | 24 | 24 | 24 |
| Expert Size | - | 512 | 512 | 512 |
| Experts per MoE Layer | – | 64 | 57 | 64 |
| FFN Experts per Token | – | 11 | – | 11 |
| Attention Experts per Token | – | – | 4 | 4 |

- *"Context: Tokyo is the capital city of Japan and has a population of over 37 million people in its metropolitan area. Question: What is the capital of Japan? The Answer is"*

UMoE successfully predicted the correct answers for both inputs, even after removing the context. For the final token in each input, we collected attention maps from the top 8 experts across all layers.

Fig. 9 presents attention maps for the final token prediction, revealing distinct patterns between higher and lower-ranked experts. Higher-ranked experts demonstrate more focused attention distributions that align with task requirements. For instance, in the Shakespeare question, Expert_0 and Expert_32 show pronounced attention weights on task-critical tokens "who wrote". Similarly, for the Tokyo question, the top expert exhibits sophisticated attention patterns by focusing on key contextual elements like "Japan" and "capital". These observations suggest that the routing mechanism effectively identifies experts capable of extracting task-relevant information through specialized attention patterns.

To create a comprehensive visualization, we aggregated attention maps of all layers by summing them, as shown in Fig. 10. The values on the left (e.g., E0: 3.83) represent the sum of router scores

Table 12: Training Hyperparameters on FineWeb-Edu and Wikitext-103.

| Hyperparameter | FineWeb-Edu | Wikitext |
|---|---|---|
| Global Batch Size | 1024 | 64 |
| Learning Rate | 4e-4 | 2.5e-4 |
| Training Steps | 50000 | 100000 |
| LR Scheduler | cosine | cosine |
| Warmup Ratio | 0.05 | 0.05 |
| GPU | H100 | H100 |

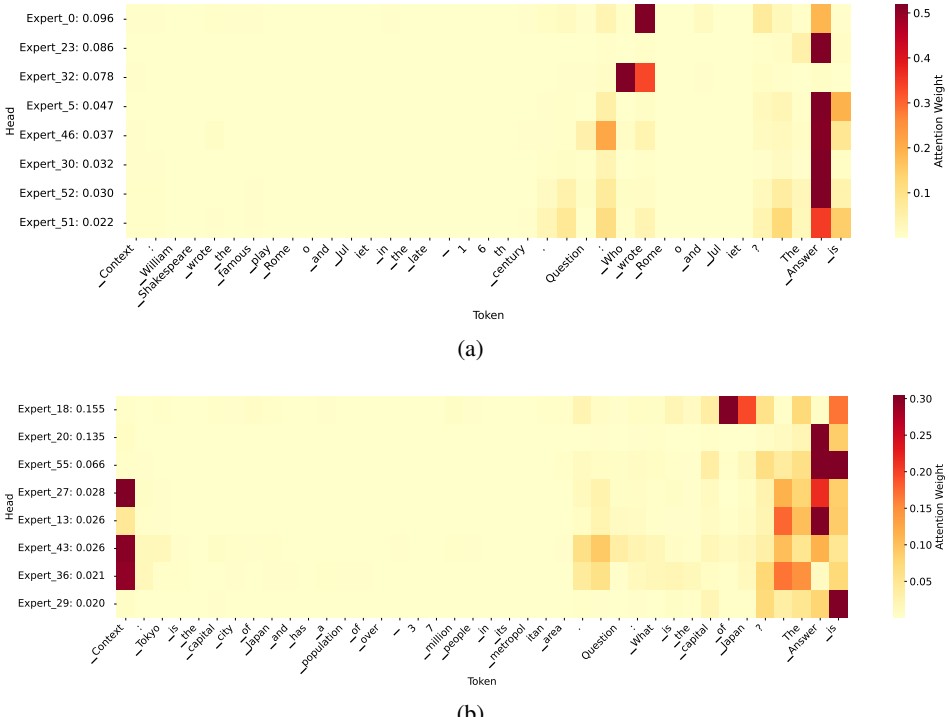

Figure 9: Representative attention maps. The heatmaps show the attention weights of the last token produced by top 8 experts, ranked by their router scores. (a) Attention patterns for the Shakespeare question, where higher-ranked experts (e.g., Expert_0, Expert_32) demonstrate focused attention on question-relevant tokens. (b) Attention patterns for the Tokyo question, showing similar task-specific attention concentration among top experts.

received by experts at each rank position across all layers. The accumulated patterns reveal that higher-ranked experts (particularly E0 and E1) maintain more targeted attention distributions focused on question-relevant tokens, while lower-ranked experts tend to focus heavily on the initial token,

Notably, we observe minimal attention paid to answer tokens present in the context. This phenomenon aligns with the conceptualization of experts (two consecutive matrices) as key-value memory modules, where input serves as a query. In other words, the output of attention layers is the composition of values, i.e., columns of the second matrix, in the activated experts, rather than token hidden states. This suggests that the token mixing should focus on building appropriate query for accurate compositions. Therefore, the last token should pay attention to the tokens relevant to the answer token, rather than the answer itself.

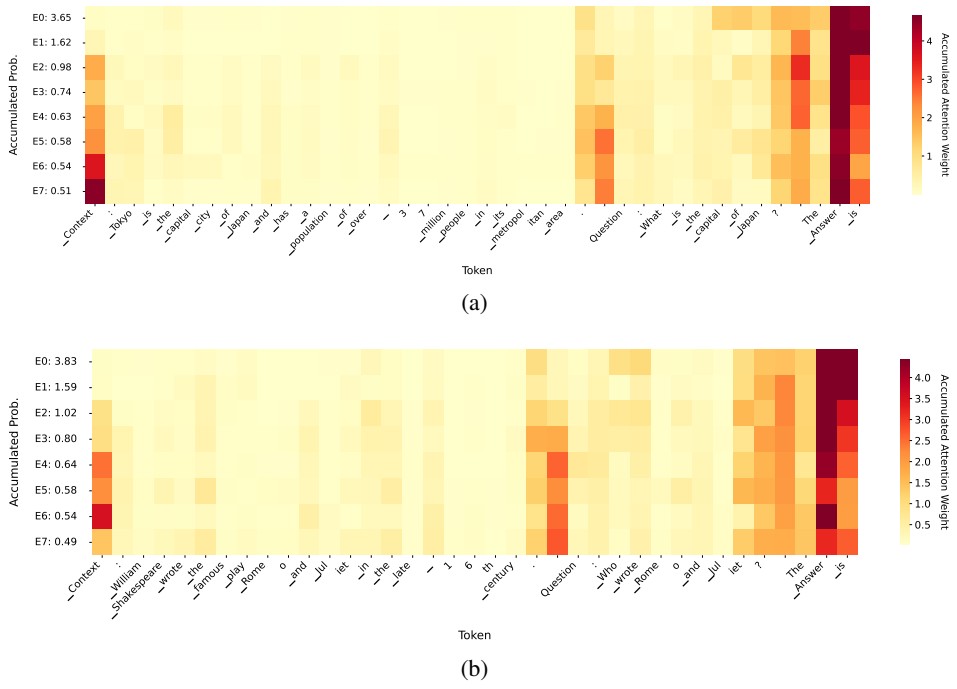

(a)

(b)

Figure 10: Layer-wise accumulated attention weights across the model. The values on the left (e.g., E0: 3.83) represent the sum of router scores for experts. Higher-ranked experts (E0, E1) consistently show more focused attention distributions compared to lower-ranked experts.

## A.6 Limitations

One limitation of UMoE is, as discussed in Section 4.2, the modest additional computational cost introduced by reformulating the attention mechanism in relatively small models. Additionally, the datasets used in this paper do not cover mathematics or code, given the scope of our work and limited computational resources. Considering recent research on MoE models' reasoning limitations, exploring how reasoning abilities scale with UMoE may provide valuable insights for future work.

