# OpenReview forum: "UMoE: Unifying Attention and FFN with Shared Experts"
_NeurIPS.cc/2025/Conference — NeurIPS 2025 spotlight_

### Official Review · Reviewer_twFT · 2025-06-10

**Clarity:** 3
**Significance:** 2
**Originality:** 3
**Rating:** 5
**Confidence:** 4

**Summary:**

This paper proposes a new MoE architecture, called UMoE. It unifies the MoE for both attention and FFN modules, and supports unified training of them using shared experts. Experiments results on efficiency and efficacy demonstrates the superiority of this method.

**Questions:**

1. Could the author present the training/inference speed (steps/s or tokens/s) for the method and the baseline?
2. Is anything missing in the head of Table 3? There are no contents in the right columns of UMoE.

**Ethical Concerns:**

["NO or VERY MINOR ethics concerns only"]

**Final Justification:**

All my concerns have been addressed and I think this work is novel and solid.

**Limitations:**

yes

**Quality:**

3

**Strengths And Weaknesses:**

### Strengths
1. The UMoE framework unifies the MoE architecture for both modules of the transformer
2. The increased expressiveness of Attention-MoE brings performance improvements for the MoE architecture


### Weaknesses
1. I observe that the performance improvements narrows when scaling the model sizes in Table 4. So I am worried about the scaling of this method .
2. Many benchmarks here could be evaluated using few-shot. So a comparison with the baselines using few-shot prompting is welcomed.

---

> ### Author Rebuttal · Authors · 2025-07-30
>
> We sincerely appreciate your thoughtful feedback and positive evaluation of our work.
>
> > **Weakness 1: I observe that the performance improvements narrows when scaling the model sizes in Table 4. So I am worried about the scaling of this method .**
>
> Thank you for this insightful observation! We acknowledge that the performance gap between UMoE and FFN-MoE narrows at a larger scale. However, UMoE remains superior to prior Attention-MoE methods. This represents a key contribution: **UMoE establishes Attention-MoE as a viable and competitive paradigm for scaling Transformers, on par with state-of-the-art FFN-MoE models**. This is a significant step forward, as previous Attention-MoE methods have struggled to match the performance of their FFN counterparts.
>
> To further investigate performance scaling, we ran a new experiment on our large model. We hypothesized that performance of UMoE could be improved by reallocating activated experts from the FFN to the attention module while keeping the total number of activated parameters constant. Due to time constraints, we trained for 10B tokens on FineWeb-Edu:
>
> | **k for Attention Layer** | **k for FFN Layer** | **Perplexity (Validation)** |
> | ------------------------- | ------------------- | --------------------------- |
> | 4                         | 11                  | 19.76                       |
> | 6                         | 9                   | 19.68                       |
> | 8                         | 7                   | **19.55**                   |
>
> This suggests that with architecture-specific tuning, the performance of UMoE at scale can be further enhanced.
>
> > **Weakness 2: Many benchmarks here could be evaluated using few-shot. So a comparison with the baselines using few-shot prompting is welcomed.**
>
> Thanks for this suggestion! As requested, we evaluated our models and the FFN-MoE baseline in a 3-shot setting using the `lm-evaluation-harness`. We found that other Attention-MoE baselines (MoA, SwitchHead) consistently underperformed FFN-MoE, so for clarity, we present a direct comparison with the strongest baseline below.
>
> | **Model**        | **Params** | **HellaSwag** | **PIQA**  | **ARC-E** | **ARC-C** | **RACE**  | **Lambada** | **MMLU** | **Wino** | **Avg.**  |
> | ---------------- | ---------- | ------------- | --------- | --------- | --------- | --------- | ----------- | -------- | -------- | --------- |
> | **Base Models**  |            |               |           |           |           |           |             |          |          |           |
> | FFN-MoE          | 535 M      | 38.48         | 67.03     | **64.78** | 31.06     | 29.48     | 22.02       | 32.1     | 52.9     | 42.23     |
> | UMoE (Att)       | 547 M      | 40.02         | **67.56** | 62.0      | **33.77** | **29.89** | 21.94       | 33.4     | 53.4     | **42.75** |
> | UMoE             | 540 M      | **41.38**     | 66.45     | 61.36     | 33.66     | 28.43     | **22.15**   | **34.0** | **53.7** | 42.64     |
> | **Large Models** |            |               |           |           |           |           |             |          |          |           |
> | FFN-MoE          | 3.8 B      | 53.38         | 74.40     | 70.08     | 41.99     | **37.99** | 35.11       | 30.4     | **57.9** | 50.16     |
> | UMoE (Att)       | 3.8 B      | 53.52         | 74.08     | 69.50     | **43.01** | 36.81     | 36.59       | 30.9     | 56.5     | 50.11     |
> | UMoE             | 3.6 B      | **54.72**     | **74.94** | **71.63** | 40.93     | 35.79     | **37.43**   | **32.8** | 57.4     | **50.70** |
>
> These few-shot results are consistent with our zero-shot findings: UMoE (Att) is highly competitive with FFN-MoE, and UMoE model demonstrates the best overall performance.
>
> > **Q1: Could the author present the training/inference speed (steps/s or tokens/s) for the method and the baseline?**
>
> This is a very constructive suggestion! We implement the forward pass of experts in MoE layers using GPU kernels written in Triton. Below are the training throughput (tokens/s) and inference latency measurements (for pre-filling 1024-token sequences) conducted on a single H100. We didn't measure token generation speed for inference since we found decoding can hardly fully utilize GPU.
>
> | **Base Model**  | **Model Size** | **Pre-filling latency (s)** | **Training throughput (token/s)** |
> | --------------- | -------------- | --------------------------- | --------------------------------- |
> | GPT             | 134 M          | **0.1376**                  | **126,508**                       |
> | FFN-MoE         | 535 M          | 0.1974                      | 74,415                            |
> | MoA             | 525 M          | 0.1843                      | 90,394                            |
> | SwitchHead      | 533 M          | 0.2253                      | 69,374                            |
> | UMoE-Att        | 547 M          | 0.2006                      | 76,538                            |
> | UMoE            | 540 M          | 0.2197                      | 71,461                            |
> | **Large Model** |                |                             |                                   |
> | GPT             | 1.1 B          | **0.2950**                  | 11,634                            |
> | FFN-MoE         | 3.8 B          | 0.4080                      | 9,799                             |
> | MoA             | 3.6 B          | 0.3624                      | **12,321**                        |
> | SwitchHead      | 3.7 B          | 0.4328                      | 8,882                             |
> | UMoE-Att        | 3.8 B          | 0.4120                      | 11,377                            |
> | UMoE            | 3.6 B          | 0.4392                      | 9,194                             |
>
> All MoE variants exhibit similar latency but require significantly more computation time than the dense baseline despite having comparable theoretical MACs. Both the routing mechanism and sparse expert computation within MoE layers introduce substantial overhead. For instance, SwitchHead and UMoE are slower than other MoE models because SwitchHead implements two MoE layers per attention head, while UMoE has two MoE layers per block (one in attention, one in FFN).
>
> Our paper's focus on MACs as the primary metric for computational efficiency, as it reflects the theoretical advantages of the architecture, abstracting away from current hardware and software implementation limitations. We believe these practical speed gaps will narrow with future advancements in GPU kernels and parallelization strategies. We will add this discussion to our paper.
>
> > **Q2: Is anything missing in the head of Table 3? There are no contents in the right columns of UMoE?**
>
> Thank you for noting this oversight! The right columns of Table 3 represent variants of our model with different parameter-sharing strategies. We apologize for the lack of clarity. We have revised the table header and caption to clearly explain each column.

---

> > ### Comment · Reviewer_twFT · 2025-08-01
> >
> > Thanks for your response.
> >
> > I still have some concerns:
> > 1. Regarding your few-shot experiments, for base model, the zero-shot performance of UMoE is slightly greater than that of UMoE (Att), but the few-shot performance of UMoE is lower than that of UMoE (Att). But this relationship is quite different when it happens to large model. I would expect the authors could explain such difference.
> > 2. After carefully reviewing the efficiency/efficacy of both baselines and your methods, I have the following concern: Dense models have much fewer parameters and much worse performance than MoE competitors. Is such a comparison fair enough?

---

> > > ### Author Response · Authors · 2025-08-01
> > >
> > > We sincerely thank the reviewer for the prompt and insightful feedback. We address the remaining concerns below.
> > >
> > > **Concern 1: The inconsistency between zero-shot and few-shot performance between UMoE and UMoE-Att across model sizes.**
> > >
> > > We hypothesize this may be attributed to two factors:
> > >
> > > 1. **Few-shot sampling variability**: To investigate this further, we conducted additional evaluations using different random seeds for sample selection. As shown in the table below, UMoE actually outperforms UMoE-Att for base models with seeds 128 and 256, though the performance gap remains smaller compared to the large models.
> > >
> > > | **Base Model** | Seed 128 | Seed 256 | Seed 512 | **Avg** |
> > > | --- | --- | --- | --- | --- |
> > > | UMoE-Att | 42.70 | 42.57 | **42.66** | 42.64 ± 0.07 |
> > > | UMoE | **42.78** | **42.68** | 42.61 | 42.69 ± 0.09 |
> > > | **Large Model** |  |  |  |  |
> > > | UMoE-Att | 50.16 | 50.08 | 50.16 | 50.13 ± 0.05 |
> > > | UMoE | **50.73** | **50.71** | **50.82** | 50.75 ± 0.06 |
> > > 1. **In-context learning capabilities of smaller models**: Prior research has shown that models with fewer than 1B parameters can demonstrate unstable few-shot learning capabilities, with performance sometimes worse than zero-shot [1]. Additionally, small models also exhibit high sensitivity to the ordering of few-shot prompts [2]. This instability may help explain why base-sized UMoE exhibits more variable performance in few-shot settings, while large-sized UMoE consistently outperforms UMoE-Att, aligning with the language modeling results.
> > >
> > > **Concern 2: Comparison with dense models.**
> > >
> > > Thank you for raising this point. We included dense models as a baseline following previous practices in MoE research [3, 4]. While dense models have fewer parameters, they represent an important comparison point because they have the same activated parameter count and theoretically similar computational overhead to the MoE models.
> > >
> > > This comparison establishes a performance baseline for models with similar inference cost and demonstrates that MoE architectures provide an alternative scaling path that increases parameter count without proportionally increasing computation overhead.
> > >
> > > [1] Language Models are Few-Shot Learners, 2020
> > >
> > > [2] Fantastically ordered prompts and where to find them: Overcoming few-shot prompt order sensitivity, ACL 2022
> > >
> > > [3] DeepSeekMoE: Towards Ultimate Expert Specialization in Mixture-of-Experts Language Models, ACL 2024
> > >
> > > [4] OLMoE: Open Mixture-of-Experts Language Models, ICLR 2025

---

> > > > ### Comment · Reviewer_twFT · 2025-08-02
> > > >
> > > > Thanks for the author's response and all my concerns have been addressed.
> > > >
> > > > I would raise my score.

---

### Official Review · Reviewer_fL6Z · 2025-06-12

**Clarity:** 3
**Significance:** 3
**Originality:** 3
**Rating:** 4
**Confidence:** 5

**Summary:**

This paper proposes a new method to combine FFN MoE and attention MoE. They reformulate the attention mechanism, propose a pre-mixing formulation and post-mixing formulation, and then apply them with MoE. The experimental results show that the pre-mixing formulation can have a better performance. The UMoE architecture achieves superior performance through attention-based MoE layers while enabling efficient parameter sharing between FFN and attention components.

**Questions:**

See weaknesses above.

**Ethical Concerns:**

["NO or VERY MINOR ethics concerns only"]

**Final Justification:**

I will keep my score.

**Limitations:**

See weaknesses above.

**Quality:**

3

**Strengths And Weaknesses:**

Pros:

- The paper is technically sound.

- The paper is easy to understand.

- The experimental results show the effectiveness of the proposed method.

Cons:

- The authors claim in line 35 that MoEs are applied on the straightforward two-matrix multiplication pattern of FFNs, which is not the case in most SOTA LLMs. They use SwiGLU instead of MLP.

- The experiments are only conducted on small LLM models with 1.1B parameters as a baseline. Models with more parameters should be used to further verify the usefulness of the proposed method.

- The inference time, memory usage and accuracies of vanilla attention pre-mixing attention and post-mixing attention without MoE should be compared with an ablation study.

---

> ### Author Rebuttal · Authors · 2025-07-30
>
> Thank you for your detailed review. We appreciate your recognition that our paper is technically sound and demonstrates the effectiveness of our proposed method.
>
> > **Weakness 1: The authors claim in line 35 that MoEs are applied on the straightforward two-matrix multiplication pattern of FFNs, which is not the case in most SOTA LLMs. They use SwiGLU instead of MLP.**
>
> Thank you for this insightful observation. We acknowledge that current SOTA LLMs predominantly use SwiGLU instead of traditional MLPs. Our focus on MLPs in the paper was primarily to highlight the structural similarity between MLPs and the ($W_v, W_o$) matrices in attention layers, which enabled us to abstract a portion of attention as an expert naturally.
>
> As illustrated in Figure 1, UMoE is designed as a framework that remains agnostic to the expert implementation. To demonstrate this flexibility, we conducted additional experiments with 10B tokens using SwiGLU as experts:
>
> | Model       | # Params | # FFN Intermediate Size | Perplexity |
> | ----------- | -------- | ----------------------- | ---------- |
> | UMoE-SwiGLU | 540 M    | 128                     | 23.565     |
> | UMoE-MLPs   | 540 M    | 192                     | **23.209** |
>
> These results indicate comparable performance, with MLP experts performing slightly better. This may be because SwiGLU requires specific hyperparameter tuning or is less effective with smaller expert sizes.
>
> > **Weakness 2: The experiments are only conducted on small LLM models with 1.1B parameters as a baseline. Models with more parameters should be used to further verify the usefulness of the proposed method.**
>
> We agree that scaling our experiments to larger models would further validate UMoE's effectiveness. Given our limited computational resources (requiring over two weeks to train models exceeding 1B parameters, with unstable resource availability), we focused on demonstrating the core concept within these constraints. Nevertheless, we believe that our findings with moderately-sized models are still significant and meaningful for the research community. UMoE presents an interesting and simple architecture that can inspire future work, and we hope this paper will encourage exploration of its scaling potential with more extensive resources.
>
> > **Weakness 3: The inference time, memory usage and accuracies of vanilla attention, pre-mixing attention and post-mixing attention without MoE should be compared with an ablation study.**
>
> Thank you for this constructive suggestion. An ablation study comparing vanilla attention with our pre-mixing and post-mixing reformulations is essential to isolate the performance gains of UMoE from the attention mechanism itself. We have run this experiment on a single H100 GPU, and the results are presented below:
>
> | GPT                  | **Perplexity** | **Pre-filling latency (s)** | **Memory footprint** |
> | -------------------- | -------------- | --------------------------- | -------------------- |
> | w/ Vanilla attention | 25.381         | 0.2823                      | 4.68 G               |
> | w/ Pre-mixing        | 25.380         | 0.2998                      | 3.93 G               |
> | w/ Post-mixing       | 25.374         | 0.3190                      | 6.18 G               |
>
> As shown, the perplexity is nearly identical across all three configurations. This is expected, as the reformulations only change the execution order of matrix multiplications within attention and should not alter the final output.
>
> Regarding efficiency, the pre-mixing attention variant shows a slightly lower latency  and memory footprint than post-mixing, making it a more efficient alternative. Furthermore, our experiments indicate that pre-mixing yields superior performance when combined with MoE architectures. Based on these findings, we selected pre-mixing as our primary configuration for UMoE.

---

> > ### Comment · Reviewer_fL6Z · 2025-08-06
> >
> > Thanks the author for the response. I will keep my score.

---

### Official Review · Reviewer_1ZC3 · 2025-07-01

**Clarity:** 2
**Significance:** 2
**Originality:** 3
**Rating:** 4
**Confidence:** 4

**Summary:**

This paper propose a new way to look at attention module of transformer architecture, and realize that part of the attention module could be viewed as a FFN expert. The author proposes to have a fully MoE-ed transformer layer -- UMoE. Through experiments, they show UMoE is able to perform better, given similar FLOPs.

**Questions:**

* Line 113, the notation is misleading --- notation doesn't agree with W_o; they should really be W_q^I
* In Table 5, does the result imply we should just always skip FFN and only have attention; this is surprising because this seems to disagree with some existing work on importance of FFN.
* Line 154, "token hidden states" is more precise; token embedding typically refers to the output of embedding matrix (even before position embedding, if any, is applied).
* What does "mixing" in "pre/post-mixing" refers to? I thought a_i X is a mixing operation.

**Ethical Concerns:**

["NO or VERY MINOR ethics concerns only"]

**Final Justification:**

My concern was addressed with additional experiments from the authors

**Limitations:**

Might be useful to make a plot of PPL-MAC to show the Pareto-frontier.

**Quality:**

3

**Strengths And Weaknesses:**

Strength:
* The attempt to move towards a fully MoE-ed layer is intriguing and interesting.
* experimentation result is thorough; but might requires more description like, in main experiment, do author have MoE layer every 2 transformer layer; or in Table 5, is the expert allocation done on all layers.

Weakness:
* important explanation missing: from Eqn 11 and Figure 3, it seems W_k and W_q are fully activated to calculate the attention, which makes the saving not as appealing as it appears when the author talks about the attention is also being sparsely activated.
* important inconsistency: Equation 11 disagree with figure 3; there doesn't seem to have a expert-specifc weight; also I have question on Equation 11: since no acitvation function is involved, how is this different from having a separate W matrix and do xW? Also, if I take Eqn 11 as it is, how could this be equivalent to a vanilla attention?
* In Table 5, UMoE's improvement over FFN-MoE seems marginal given UMoE's MAC is higher; on base model, the performance gap is clearer, but UMoE is also using relatively much more ; this might imply the method doesn't scale well. Similar observation on downstream task in Table 4. Given that MAC is used for efficiency, if the author switches to FLOPs, the improvement seems even less appealing since 1 MAC = 2 FLOPs.
* some important aspect of MoE is not addressed like how the author deals with load-balancing?

---

> ### Author Rebuttal · Authors · 2025-07-30
>
> We thank the reviewer for the detailed feedback and constructive suggestions. We are encouraged that our work was found intriguing and our experiments thorough. Below, we address each of the reviewer's points.
>
> # Experimental Details
>
> > do author have MoE layer every 2 transformer layer? in Table 5, is the expert allocation done on all layers?
>
> We apologize for omitting these details. To clarify:
>
> 1. Following previous work [1], in UMoE and all MoE baselines, every Transformer layer is replaced with one corresponding MoE layer, creating a fully MoE-based architecture.
> 2. In Table 5, the same expert allocation strategy is applied to all layers.
>
> [1] Niklas Muennighoff, Luca Soldaini, Dirk Groeneveld et al. OLMoE: Open Mixture-of-Experts Language Models, 2025. URL https://openreview.net/forum?id=xXTkbTBmqq.
>
> # Weaknesses
>
> **W1 & W2: Inconsistency between Equation 11 and Figure 3**
>
> > important explanation missing: from Eqn 11 and Figure 3...
>
> > important inconsistency: Equation 11 disagree with figure 3.
>
> We thank the reviewer for identifying this inconsistency. Figure 3 was simplified and did not accurately represent the low-rank query projections (Eqn 11). We will update the pseudo-code in Figure 3 to be fully consistent with our formulation:
>
> ```python
> # Revision for Figure 3
> # S is a sequence of token hidden states, x is the last token hidden state of S
> # For simplicity, this implementation focuses on computing the output for the last token.
>
> # Attention MoE
> K = S @ W_k
> q_shared = x @ W_q
>
> indices, probs = TopKRouter(x, k) # k is head_num
> for i, p in zip(indices, probs):
>     # Expert-specific query via low-rank adaptation
>     q_i = q_shared + x @ W_a[i] @ W_b[i]
>     # core attention
>     mixing_output = Attention(Q=q_i, K=K, V=S)
>     # Expert Computation
>     expert_output = p * Experts[i](mixing_output)
>
> # ... FFN MoE
> ```
>
> > how is this different from having a separate W matrix and do xW?
>
> Thanks for this question. The core idea behind Equation 11 is parameter efficiency. In the attention layers of UMoE, the router selects top-$k$ experts for each token. For each selected expert, a unique query vector should be generated using **expert-specific weights**. Having a separate $W_q^i \in R^{d \times d_q}$ for $i$-th expert as expert-specific weights is a solution. However, this would significantly increase parameter count.
>
> Our low-rank factorization, $W_q^i \approx W_q+W_a^iW_b^i$, is a more efficient solution. It allows each expert to have a specialized query projection by adding only the small, expert-specific low-rank matrices ($W_a^i  \in R^{d \times r}, W_b^i  \in R^{r \times d_q}$) to a shared base matrix ($W_q$).
>
> > Also, if I take Eqn 11 as it is, how could this be equivalent to a vanilla attention?
>
> Our modification affects only the query projection. The core attention mechanism—calculating attention  scores and producing a weighted sum of value vectors—remains unchanged.
>
> > it seems W_k and W_q are fully activated to calculate the attention.
>
> Yes, `W_k` and the shared `W_q` is utilized for all tokens. The computation in attention layers remains sparse. This sparsity comes from activating only a subset of experts for the remaining computations, not from the initial key/query projections.
>
> **W3: Computational Efficiency and Scaling**
>
> >  In Table 5, UMoE's improvement over FFN-MoE seems marginal given UMoE's MAC is higher; on base model, the performance gap is clearer, but UMoE is also using relatively much more; this might imply the method doesn't scale well. Similar observation on downstream task in Table 4. Given that MAC is used for efficiency, if the author switches to FLOPs, the improvement seems even less appealing since 1 MAC = 2 FLOPs.
>
> This is an insightful and crucial point! Table 2 presents a MAC-matched comparison, where UMoE consistently outperforms baselines even when they are allocated similar or greater computational budgets. This suggests UMoE's advantage stems from a more effective formulation of computation rather than simply using more computation.
>
> While the performance gap between UMoE and FFN-MoE narrows at a larger scale, UMoE remains superior to prior Attention-MoE methods. We see this as a primary contribution: **UMoE establishes Attention-MoE as a viable and competitive paradigm for scaling Transformers, on par with state-of-the-art FFN-MoE models.** This is a significant step forward, as previous Attention-MoE methods have struggled to match the performance of their FFN counterparts.
>
> To further investigate performance scaling, we ran a new experiment on our large model. We hypothesized that performance of UMoE could be improved by reallocating activated experts from the FFN to the attention module while keeping the total number of activated parameters constant. Due to time constraints, we trained for 10B tokens on FineWeb-edu.
>
> | **k for Attention Layer** | **k for FFN Layer** | **Perplexity (Validation)** |
> | ------------------------- | ------------------- | --------------------------- |
> | 4                         | 11                  | 19.76                       |
> | 6                         | 9                   | 19.68                       |
> | 8                         | 7                   | **19.55**                   |
>
> **W4: Load Balancing**
>
> > some important aspect of MoE is not addressed like how the author deals with load-balancing?
>
> Thank you for this question! UMoE employs the standard auxiliary loss for load balancing introduced by the Switch Transformer. We regret omitting this important implementation detail and will add it to our revision.
>
> # Questions
>
> > Q1: Line 113, the notation is misleading --- notation doesn't agree with W_o; they should really be W_q^I
>
> > Q3: token hidden states" is more precise.
>
> > Might be useful to make a plot of PPL-MAC to show the Pareto-frontier.
>
> We appreciate these suggestions for improving clarity and presentation. We have revised the paper accordingly and will add a PPL-MAC plot.
>
> > Q2: does the result imply we should just always skip FFN and only have attention; this is surprising because this seems to disagree with some existing work on importance of FFN.
>
> This is indeed an interesting finding we aimed to highlight — As described in Lines 141-143 and Figure 1, we can interpret a standard FFN-MoE layer as a special case of our attention formulations where the attention score is constrained to an identity matrix. On the contrary, the experts in attention layer operate on contextually-aware inputs, making them potentially more expressive. Our results in the new experiment above (W3) reinforce this, showing that shifting expert capacity to the attention module is beneficial. This is an intriguing finding that warrant further exploration.
>
> > Q4: What does "mixing" in "pre/post-mixing" refers to? I thought a_i X is a mixing operation.
>
> Yes, $a_i X$ is indeed a mixing operation. In our terminology, "mixing" refers to the weighted summation based on attention scores. We will clarify this term in the paper to avoid ambiguity.

---

> > ### Comment · Reviewer_1ZC3 · 2025-08-02
> >
> > Thanks for addressing my concern. I will raise my score

---

### Official Review · Reviewer_VfmG · 2025-07-01

**Clarity:** 3
**Significance:** 3
**Originality:** 3
**Rating:** 5
**Confidence:** 3

**Summary:**

The authors propose a novel architecture that unifies MoE design to FFN and attention layers simultaneously, termed UMoE. This can potentially allow parameter sharing between FFN and attention experts. The authors build this architecture based on the observation that FFN layers are a specialized form of attention. They also provide LLM pre-training experiments to compare the performance of the different MoEs designs matched at a similar compute budget.

**Questions:**

1. See weaknesses W1, W2, W3
2. In table 5, if attention is 0, does UMoE reduce to FFN-MoE? If not, please highlight the differences. Otherwise, this is a good insight to mention.

**Ethical Concerns:**

["NO or VERY MINOR ethics concerns only"]

**Final Justification:**

My score would have been 4.5 but decided to recommend acceptance of the paper: 5.
I think the proposed unified MoE architecture across attention and FFN layers is very interesting and novel --> opens new research directions for pre-training. This comes with good numerical performance. Additionally, throughput and memory footprint results are interesting.
This would represent a new baseline to compare against for pre-training tasks, hence my recommendation towards acceptance.
The results are interesting and potentially impactful but definitely not groundbreaking, the score shouldn't be higher.

**Quality:**

3

**Strengths And Weaknesses:**

Strengths:
1. The subject addressed in the paper is worthy of investigation. Allowing for parameter sharing between attention and FFN layers with a a strong pre-training performance can be very interesting.
2. Numerical experiments show that the author's proposed architecture can achieve a strong performance compared to SOTAs.
3. The method is well-explained. The ablation study is interesting and insightful.

Weaknesses:
1. In line 230, the authors mention that the evaluation dataset can be small and not a reliable metric for perplexity evaluation, why don't they consider another dataset, like C4?
2. Can the authors provide evaluation perplexity on other evaluation datasets to understand to what extent the performance of UMoE is generalizable?
3. Can the authors provide results of Table 2 for the larger model as well?
4. Further scaling to a ~3B dense model and other more hyperparameters for pre-training can improve the experimental results.

---

> ### Author Rebuttal · Authors · 2025-07-30
>
> We thank the reviewer for the time spent reviewing our work. We are grateful that the reviewer found our study insightful and interesting with strong performance! Below, we present the responses to the issues raised.
>
> > **Weakness 1: In line 230, the authors mention that the evaluation dataset can be small and not a reliable metric for perplexity evaluation, why don't they consider another dataset, like C4?**
>
> Thank you for this question! Our work includes both training and evaluation on two datasets, FineWeb-Edu and WikiText-103. Our primary dataset is FineWeb-Edu, which is comparable in scale to C4. FineWeb-Edu is a large-scale, diverse dataset widely used in recent language modeling research. We believe the evaluation results on FineWeb-Edu provide the most reliable insights into our model's performance.
>
> While WikiText-103 is indeed smaller, we included it primarily for fair and direct comparison with the baseline models MoA and SwitchHead, which also reported results on this dataset. The limited size of WikiText-103 helps us understand model fitting speed (as shown in Figure 5). Evaluating on additional datasets like C4 would certainly provide further insights, which we consider valuable for future work.
>
> > **Weakness 2: Can the authors provide evaluation perplexity on other evaluation datasets to understand to what extent the performance of UMoE is generalizable?**
>
> We appreciate this suggestion. To directly address the concern about generalizability, we conducted additional experiments on C4 to verify the reliability of our method. Due to computational constraints, we trained the models on 10B tokens with the following results:
>
> | Model    | Perplexity |
> | -------- | ---------- |
> | FFN-MoE  | 24.622     |
> | UMoE-Att | 24.116     |
> | UMoE     | **23.945** |
>
> Additionally, we evaluated our models trained on FineWeb-Edu with C4's validation set:
>
> | Model      | Perplexity (↓) |
> | ---------- | -------------- |
> | FFN-MoE    | 28.22          |
> | MoA        | 29.35          |
> | SwitchHead | 28.60          |
> | UMoE-Att   | 28.13          |
> | UMoE       | **27.98**      |
>
> These results confirm that our conclusions remain consistent across datasets: UMoE presents a strong alternative to previous FFN-MoE method.
>
> > **Weakness 3: Can the authors provide results of Table 2 for the larger model as well?**
>
> Thank you for pointing this out. We apologize for this oversight; the table was prepared but inadvertently omitted from the appendix. The results for the larger model are as follows:
>
> | Model      | MACs   | Active Params      | Fineweb (↓) | Wikitext (↓) |
> | ---------- | ------ | ------------------ | ----------- | ------------ |
> | FFN-MoE    | 4.81 T | 2048 + 512 x 12    | 16.00       | 25.54        |
> | MoA        | 4.91 T | 512 x 6 + 5632     | 16.61       | 26.79        |
> | SwitchHead | 4.87 T | 640 x 4 + 5632     | 16.45       | 26.12        |
> | UMoE-Att   | 4.73 T | 512 x 4 + 5632     | 16.03       | 25.53        |
> | UMoE       | 4.75 T | 512 x 4 + 512 x 11 | **15.95**   | **25.44**    |
>
> The results for the larger model are consistent with our other experiments, showing that UMoE consistently outperforms previous attention-MoE baselines even when they are allocated similar or greater computational budgets, while achieving performance on par with FFN-MoE.
>
> > **Weakness 4: Further scaling to a ~3B dense model and other more hyperparameters for pre-training can improve the experimental results.**
>
> We agree with the reviewer that scaling our experiments to larger models (~3B parameters) and exploring a wider range of hyperparameters would be valuable for further demonstrating the practical effectiveness of UMoE.
>
> Given our limited computational resources (requiring over two weeks to train models exceeding 1B parameters, with unstable resource availability), we focused on demonstrating the core concept within these constraints. Nevertheless, we believe that our findings with moderately-sized models are still significant and meaningful for the research community. UMoE presents an interesting and simple architecture that can inspire future work, and we hope this paper will encourage exploration of its scaling potential with more extensive resources.
>
> > **Question: In table 5, if attention is 0, does UMoE reduce to FFN-MoE? If not, please highlight the differences. Otherwise, this is a good insight to mention.**
>
> Thank you for this insightful question!  To clarify, when the activated expert count (k) for attention is set to 0 in Table 5, UMoE does not precisely reduce to standard FFN-MoE.
>
> In UMoE, we utilize a shared pool of experts for both attention and FFN layers, but with different numbers of activated experts (top-k) for each. If k=0 for attention, this would disable all attention mechanisms since no experts would be activated for attention computations. The model would become pure FFNs without any token interaction capabilities, functionally equivalent to a 1-gram language model predicting solely based on the current token position. Therefore, we did not train such a model configuration.

---

> > ### Comment · Reviewer_VfmG · 2025-08-04
> >
> > I appreciate the authors' thorough response and new experiments.
> > I have raised my score to 5.

---

### Official Review · Reviewer_nJ81 · 2025-07-03

**Clarity:** 2
**Significance:** 3
**Originality:** 3
**Rating:** 4
**Confidence:** 3

**Summary:**

This paper presents UMoE, a novel sparse Mixture‑of‑Experts (MoE) Transformer architecture that unifies MoE mechanisms across both attention and feed‑forward network (FFN) layers. The authors reformulate attention to reveal an FFN‑like structure, enabling shared experts between attention and FFN modules. This design supports efficient parameter sharing, eliminating the need for specialized attention MoE implementations while maintaining high performance. Experiments on language modeling (e.g., WikiText and FineWeb‑Edu) show UMoE achieves comparable or superior perplexity.

**Questions:**

See Weaknesses.

**Ethical Concerns:**

["NO or VERY MINOR ethics concerns only"]

**Final Justification:**

Thanks for the additional latency and throughput results. While the proposed UMoE achieves better perplexity at similar MACs, the increased training time and inference latency are still notable.

I believe it would be more meaningful to compare model quality under equal training throughput or inference latency. This reflects real-world constraints more accurately. Since this limitation is common in current MoE work, I will keep my score unchanged.

**Limitations:**

Yes

**Quality:**

3

**Strengths And Weaknesses:**

**Strengths**:
1. The paper introduces a novel perspective by unifying sparse MoE mechanisms in both attention and FFN layers, which simplifies model architecture.
2. By showing that attention can be reformulated as a sequence of FFNs, UMoE enables expert sharing across attention and FFN.
3. Experiments on language modeling (e.g., WikiText and FineWeb‑Edu) show UMoE achieves comparable or superior perplexity, with detailed routing analysis demonstrating expert specialization and interpretable attention behaviors.

**Weaknesses**:
1. Although the paper discusses theoretical FLOPs and parameter counts, it does not report end-to-end training/inference wall-clock time comparisons against baselines, which limits claims on practical efficiency.
2. The experimental setup is limited in scope, making it difficult to assess the generalizability of the proposed method. In particular, the paper does not evaluate UMoE in more challenging or diverse scenarios such as large-scale pretraining tasks or long-context modeling, where the benefits of unified sparse routing might be more or less pronounced. Including such experiments would strengthen the claims about the method’s broad applicability.
3. The inference performance of UMoE remains unclear and warrants further evaluation. Since language modeling relies on autoregressive generation, efficient inference typically requires KV caching. However, UMoE appears to involve caching intermediate activations (e.g., the X used in expert routing and weighted summation), which may increase both computational overhead and memory usage during inference. This overhead could be particularly problematic when compared to baselines employing optimized techniques such as Grouped Query Attention (GQA). Without benchmarking inference latency and memory footprint, it is difficult to assess UMoE’s practical viability in real-world deployment scenarios.
4. The writing quality of the paper requires proofreading and refinement.
5. The paper lacks sufficient explanation for the performance gains over other MoE-based attention methods. While empirical results are promising, the authors should provide deeper analysis—e.g., ablation studies, theoretical insights, or visualizations—to clarify why UMoE achieves better results.

---

> ### Author Rebuttal · Authors · 2025-07-30
>
> Thanks a lot for your detailed review - we are glad you found our proposed UMoE novel, which eliminates the need for specialized attention MoE implementations while maintaining high performance.
>
> Regarding your Weaknesses:
>
> > **Weakness 1: end-to-end training/inference wall-clock time comparisons against baselines**
>
> This is a very constructive suggestion! We implement the forward pass of experts in MoE layers using GPU kernels written in Triton. Below we report the training throughput (tokens/s) and inference latency (ms). Specifically, inference latency is the time required for pre-filling sequences with 1024 tokens. All measurements were conducted on a single H100.
>
> | **Base Model**  | **Model Size** | **Pre-filling latency (s)** | **Training throughput (token/s)** |
> | --------------- | -------------- | --------------------------- | --------------------------------- |
> | GPT             | 134 M          | **0.1376**                  | **126,508**                       |
> | FFN-MoE         | 535 M          | 0.1974                      | 74,415                            |
> | MoA             | 525 M          | 0.1843                      | 90,394                            |
> | SwitchHead      | 533 M          | 0.2253                      | 69,374                            |
> | UMoE-Att        | 547 M          | 0.2006                      | 76,538                            |
> | UMoE            | 540 M          | 0.2197                      | 71,461                            |
> | **Large Model** |                |                             |                                   |
> | GPT             | 1.1 B          | **0.2950**                  | 11,634                            |
> | FFN-MoE         | 3.8 B          | 0.4080                      | 9,799                             |
> | MoA             | 3.6 B          | 0.3624                      | **12,321**                        |
> | SwitchHead      | 3.7 B          | 0.4328                      | 8,882                             |
> | UMoE-Att        | 3.8 B          | 0.4120                      | 11,377                            |
> | UMoE            | 3.6 B          | 0.4392                      | 9,194                             |
>
> As the results indicate, all MoE variants show quite similar latency while requiring significantly more time than the dense baseline despite having similar theoretical MACs.
>
> Our analysis reveals that both the routing mechanism and sparse expert computation within MoE layers introduce noticeable overhead. Specifically, SwitchHead and UMoE are slower than other baseline MoE models because SwitchHead implements two MoE layers per attention head while UMoE has two MoE layers per Transformer block.
>
> These findings highlight a known challenge with MoE architectures: while theoretically efficient in computation, current hardware and software implementations don't fully realize these benefits in practice. Due to these limitations, we use MACs as the main efficiency metric in our work. We believe these practical speed gaps will narrow with future advancements in GPU kernels and parallelization strategies. We will add this discussion to our paper.
>
> > **Weakness 2: The experimental setup is limited in scope ...... In particular, the paper does not evaluate UMoE in more challenging or diverse scenarios such as large-scale pretraining tasks or long-context modeling**
>
> Thank you for this suggestion! Our paper focuses on proposing a new attention-MoE structure, following the evaluation protocols of previous related work [1, 2] by conducting pretraining on general datasets. Your suggestions regarding larger-scale training and testing on long-context scenarios are valuable and could provide new insights, yet these are beyond the scope of our current study. Since these tasks are resource-intensive, we leave them for future work.
>
> > **Weakness 3: However, UMoE appears to involve caching intermediate activations (e.g., the X used in expert routing and weighted summation), which may increase both computational overhead and memory usage during inference.**
>
> > **This overhead could be particularly problematic when compared to baselines employing optimized techniques such as Grouped Query Attention (GQA)**.
>
> > **Without benchmarking inference latency and memory footprint, it is difficult to assess UMoE’s practical viability in real-world deployment scenarios.**
>
> This is an insightful observation, especially regarding GQA! We addressed the inference latency question in W1. Here, we focus specifically on KV cache and memory footprint. The footprint is measured with ```torch.cuda.memory_allocated()```. We consider a scenario using a large model generating the next token based on 8192 tokens. All models have 4 attention heads.
>
> |             | footprint (GB) | Cache size (MB) | K Cache Shape  | V Cache Shape   |
> | ----------- | -------------- | --------------- | -------------- | --------------- |
> | FFN-MoE     | 9.75           | 1536.0 MB       | [4, 8192, 512] | [4, 8192, 512]  |
> | UMoE        | 8.21           | 960.0 MB        | [1, 8192, 512] | [1, 8192, 2048] |
> | FFN-MoE-GQA | 7.78           | 384.0 MB        | [1, 8192, 512] | [1, 8192, 512]  |
> | UMoE-MLA    | 7.74           | 384.0 MB        | [1, 8192, 512] | [1, 8192, 512]) |
>
> As shown, **UMoE has a smaller KV cache size and memory footprint compared to vanilla attention** due to our pre-mixing attention design. We acknowledge that our proposed pre-mixing attention cannot utilize Grouped Query Attention (GQA) since we already have only one key and value, which cannot be further reduced. However, we can leverage Multi-head Latent Attention (MLA) [3], a promising alternative to GQA proposed by DeepSeek. MLA reduces memory footprint by decreasing KV cache dimensions.
>
> We trained a base-size UMoE on Fineweb-Edu using 10B tokens with MLA, which resulted in only a slight degradation in performance (perplexity: 23.209 to 23.565), demonstrating MLA's effectiveness.
>
> > **Weakness 4: The writing quality of the paper requires proofreading and refinement.**
>
> We appreciate this feedback and will thoroughly revise the manuscript, addressing writing issues and incorporating suggestions from all reviewers to improve clarity and presentation.
>
> > **Weakness 5: The paper lacks sufficient explanation for the performance gains.**
>
> Thank you for this suggestion. The main distinction between MoE-based attention methods, including our UMoE, lies in expert design—specifically how to increase attention parameters most effectively. One possible explanation for our superior performance is that the two-matrix multiplication pattern in the attention layer resembles memory layers (as mentioned in Section 3.3). This suggests that matrices $W_v$ and $W_o$ should be scaled together, which previous work failed to accomplish. MoA didn't scale them together, and both MoA and SwitchHead break key-value correspondence during scaling. In the revised manuscript, we will expand Section 3.3 to provide a more comprehensive discussion.
>
> [1] Csordás, R., Piękos, P., Irie, K., and Schmidhuber, J. SwitchHead: Accelerating Transformers with Mixture-of-Experts Attention, NeurIPS 2024. https://openreview.net/forum?id=80SSl69GAz
>
> [2] Zhang, X., Shen, Y., Huang, Z., Zhou, J., Rong, W., and Xiong, Z. Mixture of Attention Heads: Selecting Attention Heads Per Token, EMNLP 2022. https://aclanthology.org/2022.emnlp-main.278/
>
> [3] DeepSeek-AI et al. DeepSeek-V2: A Strong, Economical, and Efficient Mixture-of-Experts Language Model, 2024. https://arxiv.org/abs/2405.04434

---

> > ### Comment · Reviewer_nJ81 · 2025-08-06
> >
> > Thanks for the additional latency and throughput results. While the proposed UMoE achieves better perplexity at similar MACs, the increased training time and inference latency are still notable.
> >
> >
> > I believe it would be more meaningful to compare model quality under equal training throughput or inference latency. This reflects real-world constraints more accurately. Since this limitation is common in current MoE work, I will keep my score unchanged.

---

### Note · Authors · 2025-08-11

We sincerely thank the reviewers and area chairs for their time, effort, and insightful feedback. We are encouraged by the positive assessments, noting the novelty of our unified MoE architecture across attention and FFN layers (nJ81, VfmG, 1ZC3), thorough empirical evaluation (1ZC3), insightful ablations (VfmG), strong empirical performance (nJ81, VfmG, fL6Z, twFT), and clarity of presentation (fL6Z, VfmG).

**Rebuttal Summary**

We are delighted that our detailed responses and new experiments were well-received, leading to a consensus of positive ratings among the reviewers.

Due to the space limit, we highlight only the main shared concerns addressed in our rebuttal and discussion:

- **Wall-clock time measurements (nJ81, twFT)** — To strengthen our work, we measured training throughput and inference latency for UMoE and all baselines. UMoE has latency comparable to other MoE variants, with overhead arising primarily from the inherent routing and sparse computation costs of MoE, rather than from our architectural design.
- **Scaling behavior (twFT, 1ZC3)** — While the gap with FFN-MoE narrows at larger scales, UMoE consistently outperforms prior Attention-MoE methods and remains slightly better than FFN-MoE. This is a significant step forward, as previous Attention-MoE methods have struggled to match the performance of their FFN counterparts. We also show that UMoE can achieve further improvements under the same parameter budget via reallocating experts from FFN to attention layers.

- **Memory usage (nJ81, fL6Z)** — Reviewers expressed interest in the memory footprint of our proposed pre-mixing attention, particularly its impact on the KV cache. We show that pre-mixing attention reduces KV cache size compared to vanilla attention; combined with Multi-head Latent Attention (MLA), memory usage can be further reduced.
- **Further scaling (VfmG, fL6Z)** — We agree that scaling beyond 3B parameters would be informative, but it is resource-intensive and beyond our current resources. We believe our results on moderately-sized models already offer substantial and meaningful evidence of UMoE’s effectiveness. We hope this paper will inspire future exploration of its scaling potential with more extensive resources.

We again thank all reviewers for their thoughtful and constructive feedback, which has been highly valuable in strengthening our work. We have carefully considered both the shared concerns and each reviewer’s individual points in revising our paper.

---

### Decision · Program_Chairs · 2025-09-17

**Decision:**

Accept (spotlight)

**Comment:**

This paper argues that part of attention block of transformer architecture can be viewed as FFN expert. With this insight, authors propose a full mixture-of-expert transformer layer - UMoE. Experiments indicate that given similar compute, UMoE achieves superior performance and enables efficient parameter sharing between FFN and attention.

All the reviewers commend novel insights and proposed method that enables expert sharing across attention and FFN. The paper is well written and easy to follow and the contributions are clear.  Empirical results are compelling, however authors are encouraged to include additional details wrt experiment setup and analysis of the experimental results, including additional ablations.

Authors did a great job during the rebuttal phase, providing additional evaluations requested by the reviewers. We expect these new results and explanations to be incorporated into the final version of the paper.

Additionally reviewers encourage further proofreading of the paper